# The Potential Role of Peripheral Oxidative Stress on the Neurovascular Unit in Amyotrophic Lateral Sclerosis Pathogenesis: A Preliminary Report from Human and In Vitro Evaluations

**DOI:** 10.3390/biomedicines10030691

**Published:** 2022-03-17

**Authors:** Elena Grossini, Divya Garhwal, Sakthipriyan Venkatesan, Daniela Ferrante, Angelica Mele, Massimo Saraceno, Ada Scognamiglio, Jessica Mandrioli, Amedeo Amedei, Fabiola De Marchi, Letizia Mazzini

**Affiliations:** 1Laboratory of Physiology, Department of Translational Medicine, University East Piedmont, 28100 Novara, Italy; elena.grossini@med.uniupo.it (E.G.); 20029627@studenti.uniupo.it (D.G.); sakthipriyan.venkatesan@uniupo.it (S.V.); 2Statistic Unit, Department of Translational Medicine, University East Piedmont, 28100 Novara, Italy; daniela.ferrante@med.uniupo.it; 3ALS Center, Neurology Unit, Department of Translational Medicine, University East Piedmont, 28100 Novara, Italy; 20011892@studenti.uniupo.it (A.M.); 20031957@studenti.uniupo.it (M.S.); adascognamiglio08@gmail.com (A.S.); fabiola.demarchi@uniupo.it (F.D.M.); 4Department of Biomedical, Metabolic and Neural Sciences, University of Modena and Reggio Emilia, 41125 Modena, Italy; mandrioli.jessica@aou.mo.it; 5Neurology Unit, Azienda Ospedaliero Universitaria di Modena, 41126 Modena, Italy; 6Department of Experimental and Clinical Medicine, University of Florence, 50134 Florence, Italy; amedeo.amedei@unifi.it

**Keywords:** astrocytes, endothelial cells, mitochondria, nitric oxide, oxidative stress

## Abstract

Oxidative stress, the alteration of mitochondrial function, and changes in the neurovascular unit (NVU) could play a role in Amyotrophic Lateral Sclerosis (ALS) pathogenesis. Our aim was to analyze the plasma redox system and nitric oxide (NO) in 25 ALS new-diagnosed patients and five healthy controls and the effects of plasma on the peroxidation/mitochondrial function in human umbilical cord-derived endothelial vascular cells (HUVEC) and astrocytes. In plasma, thiobarbituric acid reactive substances (TBARS), glutathione (GSH), and nitric oxide (NO) were analyzed by using specific assays. In HUVEC/astrocytes, the effects of plasma on the release of mitochondrial reactive oxygen species (mitoROS) and NO, viability, and mitochondrial membrane potential were investigated. In the plasma of ALS patients, an increase in TBARS and a reduction in GSH and NO were found. In HUVEC/astrocytes treated with a plasma of ALS patients, mitoROS increased, whereas cell viability and mitochondrial membrane potential decreased. Our results show that oxidative stress and NVU play a central role in ALS and suggest that unknown plasma factors could be involved in the disease pathogenesis. Quantifiable changes in ALS plasma related to redox state alterations can possibly be used for early diagnosis.

## 1. Introduction

Amyotrophic Lateral Sclerosis (ALS) is a neurodegenerative, rare, and fatal disease characterized by the selective loss of the upper and lower motor neurons (MNs). The disease comprises a predominant sporadic form and a familial variant, affecting up to 10–15% of cases. About 20% of patients with familial ALS have inherited superoxide dismutase-1 (SOD1) mutations. Various pathophysiological mechanisms have been involved in explaining the progressive degeneration of MNs, among which are oxidative stress, mitochondrial abnormalities, glutamate-related excitotoxicity, impaired responses to hypoxia, and neuroinflammation [1,2].

More recently, it has been hypothesized that changes of the neurovascular unit (NVU), composed of vascular cells, glial cells, and neurons, play a role in ALS pathogenesis as well. The disruption of NVU could hamper the blood-brain barrier (BBB) or blood-spinal cord barrier (BSCB), which would lead, in turn, to MNs damage due to the harmful effects of unknown circulating factors entering the central nervous system (CNS) and/or to changes in vascular blood flow [3,4,5]. Indeed, studies in animal models and ALS patients have shown the degeneration of endothelial cells and astrocytes end-feet processes surrounding microvessels [6,7]. In addition, BSCB alterations were reported in SOD1 mutant mice and rats before MNs degeneration, which would suggest that vascular dysfunction could represent an early pathogenic event in ALS [8].

ALS therapy aims to prevent the progression of symptoms and reverse the loss of MNs by the modulation of oxidative stress and excitotoxicity. To date, only riluzole, with an anti-glutamatergic action [9], and edaravone, a free radical scavenger, have been approved by the Food and Drug Administration (FDA) as therapeutic agents for ALS treatment [10]; however, they have shown only limited benefits in slowing the disease progression. Therefore, the advances in the knowledge of pathogenic mechanisms of ALS are relevant for finding new experimental disease-modifying treatments. Moreover, the discovery of any circulating blood biomarker that is easily quantifiable could have interesting implications for the diagnosis, prognosis, and monitoring of ALS patients.

This study aimed to examine the plasma redox state in ALS patients, and to analyze the effects of plasma on cell viability, mitochondrial membrane potential, nitric oxide (NO), reactive oxygen species (ROS), and mitochondrial ROS (mitoROS) release by members of the NVU unit, such as endothelial cells and astrocytes.

## 2. Materials and Methods

### 2.1. Patients

The study was conducted on 25 consecutive patients diagnosed with ALS defined according to the El-Escorial criteria [11] at the ALS Tertiary Center at the “Maggiore della Carità University Hospital”, Novara, Italy, between September 2020 and May 2021. The comparison was performed with a control group of subjects (*n* = 5) collected from unrelated healthy patients’ caregivers and staff members of the Department of Physiology. The study was executed in compliance with Good Clinical Practice guidelines and the ethical principles of the Declaration of Helsinki. The Hospital Ethical Committee also approved it (CE 54/17); each participant gave written informed consent for the handling of clinical data and the use of plasma samples for experimental purposes.

We recruited patients with the following inclusion criteria: age between 18 and 75 years; defined, clinically probable and probable-laboratory supported ALS based on El Escorial Criteria [11]; within 24 months from symptoms onset; patients who had not yet started any modifying therapies for ALS (riluzole/edaravone); patients without interfering comorbidities that can generate bias (e.g., other neurological, oncological, autoimmune diseases); patients able and willing to provide informed consent or who have had a legally authorized representative willing to do so.

For each patient, we collected demographic and clinical variables, such as age at onset, sex, site of onset, date of diagnosis, and diagnostic delay for each patient. We also obtained ALS Functional Rating Scale—Revised (ALSFRS-R) scores, Forced Vital Capacity percentages (FVC%), Body Mass Index (BMI) and mutational status (including c9orf72, SOD1, TARDBP and FUS). Data were collected from clinical records using an anonymous data abstraction form and an alphanumeric code was randomly assigned to each patient.

### 2.2. Blood Sample Collection

In each subject, blood samples were taken at 9 am in fasting conditions by using specific tubes (BD Vacutainer; sodium heparin as anticoagulant). Each sample was forthwidth centrifuged for 10 min through a centrifuge (Eppendorf, mod. 5702 with rotor A-4-38) working at 3100 rpm, 4 °C. The plasma was then aliquoted into 5 tubes which were stored at −80 °C at the Physiology laboratory, University East Piedmont. Those samples were used for the quantification of markers of redox state and to perform the in vitro experiments on human vascular endothelial cells (HUVEC) and astrocytes. Plasma samples were handled in pseudonymized conditions.

### 2.3. Plasmatic Quantifications

#### 2.3.1. Analysis of Plasma Thiobarbituric Acid Reactive Substances (TBARS)

The evaluation of plasma TBARS was performed on each plasma sample using the TBARS assay Kit (Cayman Chemical, Ann Arbor, MI, USA) which evaluates the malonyldialdeide (MDA) release [12]. To do this, 100 µL of each plasma sample was added to sodium dodecyl sulfate solution (100 µL) and Color Reagent (2 mL). Each sample was boiled for 1 h and then shifted on ice for 10 min to stop the reaction. Thereafter, each sample was subjected to centrifugation (10 min at 1600× *g*, 4 °C); after that, 150 μL was transferred to 96-well plates for MDA detection through a spectrophotometer (VICTOR™ X Multilabel Plate Reader; PerkinElmer; Waltham, MA, USA) at excitation/emission wavelengths of 530–540 nm. In order to quantify the correct value of TBARS in each sample (expressed as MDA in µM), a reference standard curve with the TBARS Standard was prepared. Each measurement was performed in triplicate.

#### 2.3.2. Plasma Glutathione (GSH) Quantification

GSH measurement was performed on all plasma samples through the Glutathione Assay Kit (Cayman Chemical, Ann Arbor, MI, USA) [12]. Briefly, each plasma sample was deproteinated by adding meta phosphoric acid solution in an equal volume. After centrifugation at 2000 *g* for 2 min, the supernatant of each sample was collected and 50 μL/mL of TEAM reagent was added to increase the pH. A total of 50 μL of the samples was moved to 96-well plates where the measurement of GSH was executed through a spectrophotometer (VICTOR™ X Multilabel Plate Reader) using excitation/emission wavelengths of 405–414 nm. To perform an accurate GSH quantification (as µM), a reference curve was prepared by using the GSH Standard. Each measurement was performed in triplicate.

#### 2.3.3. Plasma NO Quantification

NO was detected in all plasma samples in the form of plasma nitrite levels by the Griess system (Promega Italia Srl, Milan, Italy). To do this, 5 mL plasma sample was deproteinated by adding 10 mL sulfosalicylic acid. After that, the samples were subjected to vortex every 5 min and left to react for 30 min at room temperature. After centrifugation at 10,000× *g* for 15 min, 50 μL of the supernatant was added to saline (100 μL; 1:2 dilution) for subsequent analysis. The remaining microliters were used without dilution. To reduce nitrate to nitrite, the samples were passed through the copper–cadmium column of an autoanalyzer (Autoanalyzer; Technicon Instruments Corp., Tarrytown, NY, USA), and then they were mixed with an equal volume of Griess reagents. After 10 min, the absorbance was measured by a spectrometer (VICTOR™ X Multilabel Plate Reader, Waltham, MA, USA) at 570 nm and the NO release was examined in comparison with standard curve and given as nitrite production (μM) [13]. Each measurement was performed in triplicate.

### 2.4. In Vitro Experiments

#### 2.4.1. Effects of Plasma Samples

For the in vitro experiments on cell viability, mitochondrial membrane potential, and mitochondrial ROS release (mitoROS), the plasma of five randomly selected patients and of five controls was used. Instead, for NO release measurements, the experiments were performed with plasma taken from 10 randomly selected patients. Untreated cells were also included in the analyses.

In order to examine the responses of HUVEC cells and astrocytes treated with plasma samples in terms of cell viability (MTT procedure), mitochondrial membrane potential (JC-1 procedure), mitochondrial ROS (mitoROS; MitoROS assay), and NO release (Griess assay), specific Transwell inserts were used (Figure 1).

These inserts are permeable supports that allow the uptake and secretion of molecules by cells positioned on both their basal and apical surfaces and, thus, they achieve the analysis of cellular cross-talk in a more natural fashion. To do this, 10% plasma samples calculated in relation with total volume of each insert were positioned in the apical surface of the insert itself for 3 h, whereas HUVEC cells or astrocytes were plated in the basal one. After 3 h stimulation with plasma, the inserts were removed and various assays were performed as described below, and analyses were executed using a spectrophotometer.

#### 2.4.2. Cell Cultures

HUVEC were taken from ATCC (catalog. no. CRL-1730TM), and immortalized mouse astrocytes from hippocampi were kindly provided by prof. Dmitry Lim [14]. HUVEC and astrocytes were cultured in Dulbecco’s modified Eagle’s medium (Sigma-Aldrich, Milan Italy) enriched with 10% fetal bovine serum (FBS; Euroclone, S.p.A.; Pero, Milan, Italy), 2 mM L-glutamine (Euroclone), and 1% penicillin/streptomycin (Sigma-Aldrich, Milan, Italy).

#### 2.4.3. Cell Viability

The viability of HUVEC and astrocytes was evaluated by 1% 3-[4,5-dimethylthiazol-2-yl]-2,5-diphenyl tetrazolium bromide (MTT; Life Technologies Italia, Monza, Italy) dye [15]. To do this, 50,000 HUVEC/astrocytes/well were plated in 24-Transwells plates in complete medium (DMEM supplemented with 10% FBS). HUVEC and astrocytes were treated with 10% plasma for 3 h, as described in the experimental protocol.

After each stimulation, the medium was replaced with fresh culture medium with 0% red phenol and 0% FBS. MTT dye was added to the well plates containing the cells and left in an incubator for 2 h at 37 °C. After that, the medium was replaced with a MTT solubilization solution (dimethyl sulfoxide; Sigma, Milan, Italy) and mixed until the complete dissolution of formazan crystals. The viability of HUVEC/astrocytes was evaluated by measuring the absorbance at 570 nm through a spectrometer (VICTOR™ X Multilabel Plate Reader). Control cells (not treated cells) were set as 100% viability. Experiments have been executed in triplicate and repeated three times on different pools of HUVEC and astrocytes.

#### 2.4.4. Mitochondrial Membrane Potential Measurement

In HUVEC/astrocytes, mitochondrial membrane potential was examined through the JC-1 procedure. Briefly, 50,000 HUVEC/astrocytes/well positioned in 24-Transwells plates in complete medium were treated following the same procedures as described for MTT assay. After 3 h stimulation with plasma, the medium of HUVEC/astyrocytes plated in starvation medium was taken away and cells were incubated for 15 min at 37 °C with 5,51,6,61-tetrachloro-1,11,3,31 tetraethylbenzimidazolyl carbocyanine iodide (JC-1) 1X diluted in Assay Buffer 1X (Cayman Chemical, Ann Arbor, MI, USA) [15]. After that, the cells were washed twice by using the Assay Buffer 1X and the mitochondrial membrane potential was quantified by reading the red (excitation 550 nm/emission 600 nm) and green (excitation 485 nm/emission 535 nm) fluorescence through a spectrometer (VICTOR™ X Multilabel Plate Reader; PerkinElmer). Normalization of the data was executed versus untreated cells (control cells). Experiments were executed in triplicate and repeated three times on different pools of HUVEC and astrocytes.

#### 2.4.5. MitoROS Release

The Cayman’s Mitochondrial ROS Detection Assay Kit (Cayman Chemical, Ann Arbor, MI, USA) was used to analyze the mitoROS production [16]. To do this, 50,000 HUVEC/astrocytes/well were positioned in 24-Transwells plates in complete medium. The experimental procedures were the same as those used for MTT and JC-1 determinations. After each treatment, the reactions were blocked by replacing the culture media with 120 µL of cell-based assay buffer. After that, the buffer was aspirated and 100 µL of Mitochondrial ROS Detection Reagent Staining Solution, was added in each well and incubated at 37 °C, protected from light for 20 min. After this time, the staining solution was removed and each well was washed with 120 µL of PBS for three times. The release of mitoROS was analyzed by reading the excitation and emission wavelength at 480 nm and 560 nm, respectively, through a spectrophotometer (VICTOR™ X Multilabel Plate Reader). Normalization of the data was executed versus untreated cells (control cells) Experiments were performed in triplicate and repeated three times on different pools of HUVEC and astrocytes.

#### 2.4.6. NO Release

In HUVEC and astrocytes, NO release was evaluated through the use of the Griess method (Promega) [15]. For the experiments, 50,000 HUVEC/astrocytes for well/insert were plated in 24-Transwells plates in complete medium, and then the same experimental protocol followed for MTT, JC-1 and mitoROS methods was used. At the end of the stimulations, NO production in the sample’s supernatants was examined by adding an equal volume of Griess reagent following the manufacturer’s instruction. The reading of each sample was performed at 570 nm through a spectrometer (VICTOR™ X Multilabel Plate Reader). A standard curve was prepared to quantify the NO production, which was expressed as nitrites (μM). Experiments were executed in triplicate and repeated three times on different pools of HUVEC and astrocytes.

### 2.5. Statistical Analysis

All data were collected using the Research Electronic Data Capture software (REDCap, Vanderbilt University, Nashville, TN, USA). The mean of the multiple measurements taken for each patient was considered for the analysis. Quantitative variables are presented as median and interquartile range (IQR) for cases and controls. We evaluated the differences between two groups through the Mann—Whitney test. Spearman’s correlation coefficient was used to calculate the correlation between quantitative variables. A *p*-value < 0.05 was considered statistically significant. Statistical analysis was performed using STATA version 16 for Microsoft Windows (SAS Institute Inc., Cary, NC, USA) and Graph PAD (GraphPad Software, San Diego, CA, USA).

## 3. Results

### 3.1. Patients

ALS patients’ demographic and phenotypic data are described in Table 1. Healthy subjects were 3 males and 2 females and the age ranged from 53 years to 65 years. No significant differences were observed in sex and age between patients and controls (*p* > 0.05).

### 3.2. Plasmatic Quantifications

As shown in Figure 2, plasma TBARS levels of ALS patients were higher than those of controls (median TBARS respectively of 13.5 IQR 10.5–17.4 μM and 3 IQR 2.3–4 μM; *p* = 0.0005) (Panel A). In addition, plasma GSH levels were lower in all ALS patients than controls (median GSH respectively of 2.5 IQR 1.1–2.9 μM and 4.6 IQR 4.5–4.6 μM; *p* = 0.0005) (Panel B). Furthermore, ALS plasma median NO levels were lower than those found in controls, amounting to 4.5 IQR 1.5–5.3 μM vs. 12.4 IQR 12–13 μM (*p* = 0.0009) (Panel C). No statistically significant correlation was found between clinicdemographic features (i.e., age, sex, site of onset, ALSFRS, R, FVC%, and BMI at baseline) and plasmatic TBARS, GSH, and NO.

### 3.3. In Vitro Experiments

For in vitro experiments, we analyzed plasma from five patients and five controls for cell viability, mitochondrial membrane potential, and ROS release evaluation. Instead, the NO release was examined in both HUVEC and astrocytes treated with plasma of 10 patients. In HUVEC treated with ALS patient plasma, both cell viability and mitochondrial membrane potential were reduced (Figure 3A,B; *p* = 0.008). In addition, NO release was also lower when HUVEC were treated with the plasma of ALS patients (Figure 3D; *p* = 0.008), whereas mitoROS was higher (Figure 3C; *p* = 0.008). On the contrary, control plasma did not exert any effect on HUVEC (Figure 3).

Similar findings obtained in HUVEC were observed in astrocytes. Hence, both cell viability and mitochondrial membrane potential were reduced by the treatment of ALS plasma (Figure 4A,B; *p* = 0.008), whereas mitoROS release was higher (Figure 4C; *p* = 0.008). Regarding NO release, a change in response to ALS plasma was found in astrocytes (*p* = 0.03) (Figure 4D). Control plasma did not elicit any effect on astrocytes (Figure 4).

## 4. Discussion

The results of our study document the alteration of the redox state in the plasma of ALS patients, and the harmful effects of their plasma on NVU members.

The study is based on the hypothesis that any circulating factor(s) could play a role in the disease pathogenesis by affecting NVU. Indeed, functional and structural alterations in the NVU could represent the pathogenic event triggering the BBB changes observed in animal models and ALS patients [5].

The obtained findings focus on the main actors involved in ALS pathogenesis: oxidative stress and inflammation on the one side and vascular endothelial cells and astrocytes on the other [17,18]. The reactive astrocytes and microglia, peripheral immune cells infiltration, and elevated inflammatory mediators have widely been described in motor regions of the CNS [19]. In addition, high levels of proinflammatory cytokines such as tumor necrosis factor alpha (TNF-α), interleukin (IL) 1 beta, and interferon gamma (IFN-γ) have been identified in the cerebrospinal fluid (CSF) of ALS patients [20], and an increased amount of IL-6 was observed in astrocyte-derived exosomes of patients suffering from ALS [21].

Moreover, oxidative stress, which is related to the increased production of ROS accompanied by the reduction of the antioxidant systems, may contribute to the exacerbation of ALS progression by affecting NVU and causing the degeneration of neuromuscular junction and the decline of acetylcholine release or content in the synaptic space. Previous evidence showed increased oxidative damage in neuronal postmortem tissues, measured as MDA-modified proteins 8 hydroxy-deoxyguanosine, and nitrotyrosine products [22,23].

In our study, plasma levels of TBARS, which are markers of lipid peroxidation [24], were increased and, in contrast, those of GSH were decreased in ALS patients, independently of the demographic, clinical, and genetic features of patients. These data are consistent with previous ones showing that the response to oxidative stress is dampened in ALS. Hence, GSH levels were reduced in the motor cortex of patients compared with controls, whereas circulating serum protein carbonyls, which provide a surrogate marker for oxidative stress, were increased in ALS [25]. In addition, ALS erythrocytes showed increased lipid peroxidation as well as a reduction of antioxidants enzymes like catalase and glutathione reductase, and of GSH, which was related to the disease duration [26].

In addition, our study aimed to examine the role of any unknown circulating factor(s) on cell viability, oxidants’ release, and mitochondria function in both HUVEC and astrocytes. The treatment of HUVEC and astrocytes with the plasma of ALS patients was able to decrease cell viability and cause damage to mitochondrial function, measured as the mitochondrial membrane potential. It should be pointed out that mitochondrial dysfunction has been widely described as a pathology hallmark of ALS [27]. Hence, mitochondria are one of the most important sites for ROS production due to their role in ATP synthesis [28]. In particular, superoxide radical anion (O^−2^) is one of the main ROS released by the mitochondrial electron chain transport [29], particularly in complexes I and III. Furthermore, the free radical and high reactive O^−2^, which is considered the precursor of mitochondrial H_2_O_2_, can quickly cross the mitochondrial membranes irrespective of the energization level and give origin to the very high reactive hydroxyl radical (HO•) [30,31].

Changes in mitochondria morphology have been reported in the neurons, glial cells, and muscle cells of SOD1 [32] and C9orf72-mutated patients [33] and SOD1G93A and TDP-43A315T mice [34,35,36]. Those alterations could lead to cascading events affecting mitochondrial respiration and ATP production, and finally, cause an increase in oxidative stress [28].

In line with those issues, in our study, the reduction of mitochondrial membrane potential was accompanied by augmented ROS release by both HUVEC and astrocytes treated with the plasma of ALS patients. Furthermore, those effects were accompanied by a reduction in the cell viability by both cell types and a reduction of NO release by HUVEC.

Our finding of an alteration of mitochondrial membrane potential is relevant since this status in physiologic conditions regulates the mitochondrial permeability transition pore (mtPTP) opening. The fall of mitochondrial membrane potential observed in oxidative stress conditions is accompanied by the mtPTP opening, which can cause mitochondrial swelling and apoptosis activation [37].

In this way, it could be hypothesized that at the base of the reduced cell viability we found in HUVEC and astrocytes, there could be alterations of mtPTP and the starting of the apoptotic process.

The data we obtained regarding astrocytes agree with previous findings of the role of astrocytes in the neurodegenerative mechanisms at the ALS origin [19]. Therefore, the loss of normal astrocytic function could be at the base of the neurodegeneration in ALS and may represent a target for the development of therapies aimed at reducing oxidative stress in patients’ astrocytes and motor neurons.

Regarding the vascular endothelial component, its role in the ALS genesis has not yet been well investigated. The results about plasma NO levels and in vitro from HUVEC could add information on this issue. Hence, not only were plasma NO levels reduced in ALS patients, the plasma of ALS patients was also able to reduce NO release by HUVEC.

In astrocytes treated with ALS plasma, we also found a reduction in NO release, which is in disagreement with previous data showing an increased NO production and/or NO synthases expression/activation [38,39,40]. This discrepancy could be related to the different timing of our cell stimulation, which was limited to only 3 h instead of 8 h or even more. Future experiments could be planned to better address this issue.

In addition to the limitation of this study associated with the measurement of NO in astrocytes, another limit is represented by not having discriminated against the nature of the circulating factor(s) responsible for cellular effects. Hence, whole plasma can not target any specific circulating element(s) that could affect the toxicity towards HUVEC and astrocytes. As reported above, it could thus be hypothesized that unknown circulating factor(s) could hamper astrocytes and endothelial cell viability and mitochondria function, which, in turn, could represent a trigger event playing a pathogenic role in the ALS onset. Functional alterations in the BBB or BSBC in patients and animal models of disease have been reported as starting pathological triggers [6]. In particular, previous studies aiming to examine ALS as a neurovascular disease have highlighted the degeneration of cell members of the NVU [41]. Thus, future studies could be planned with the aim of analyzing in more detail the nature of circulating factor(s) and their role in the changes of the permeability of the BBB and BSCB, which could be evaluated through specific experimental methods. Moreover, the use of ROS scavengers could corroborate our results about oxidative stress and open new research perspectives.

It should be highlighted that the aspect related to the presence of unidentified marker(s) circulating in the plasma of ALS patients could have important clinical implications in the diagnosis and follow-up of patients. In addition, it could represent a field of application for new therapeutic approaches. Furthermore, the findings obtained in vitro through the use of ROS scavengers could have implications in the clinical setting relating to the possible development of new treatments for ALS.

We found no statistically significant correlation between plasmatic results and clinical variables, which may be due to the small sample of patients, representing a main limitation of the study. In addition, this was an explorative study deserving further investigation since patients’ follow-up is still ongoing, and we cannot conclude from a prognostic point of view. A new and larger study could contribute to a better understanding of the clinical significance of our promising findings.

## 5. Conclusions

In conclusion, we established a plasmatic and in vitro model on the redox state in ALS patients, in agreement with what is already known in the literature, showing a fundamental role of oxidative stress, the alteration of mitochondrial function, and NVU in ALS pathogenesis. Quantifiable plasma changes related to the redox state can possibly be used for early ALS diagnosis.

## Figures and Tables

**Figure 1 biomedicines-10-00691-f001:**
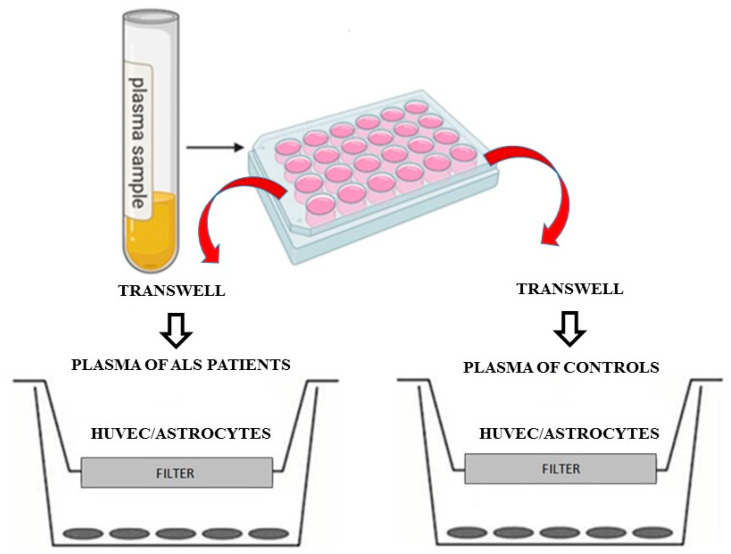
In vitro experimental protocol.

**Figure 2 biomedicines-10-00691-f002:**
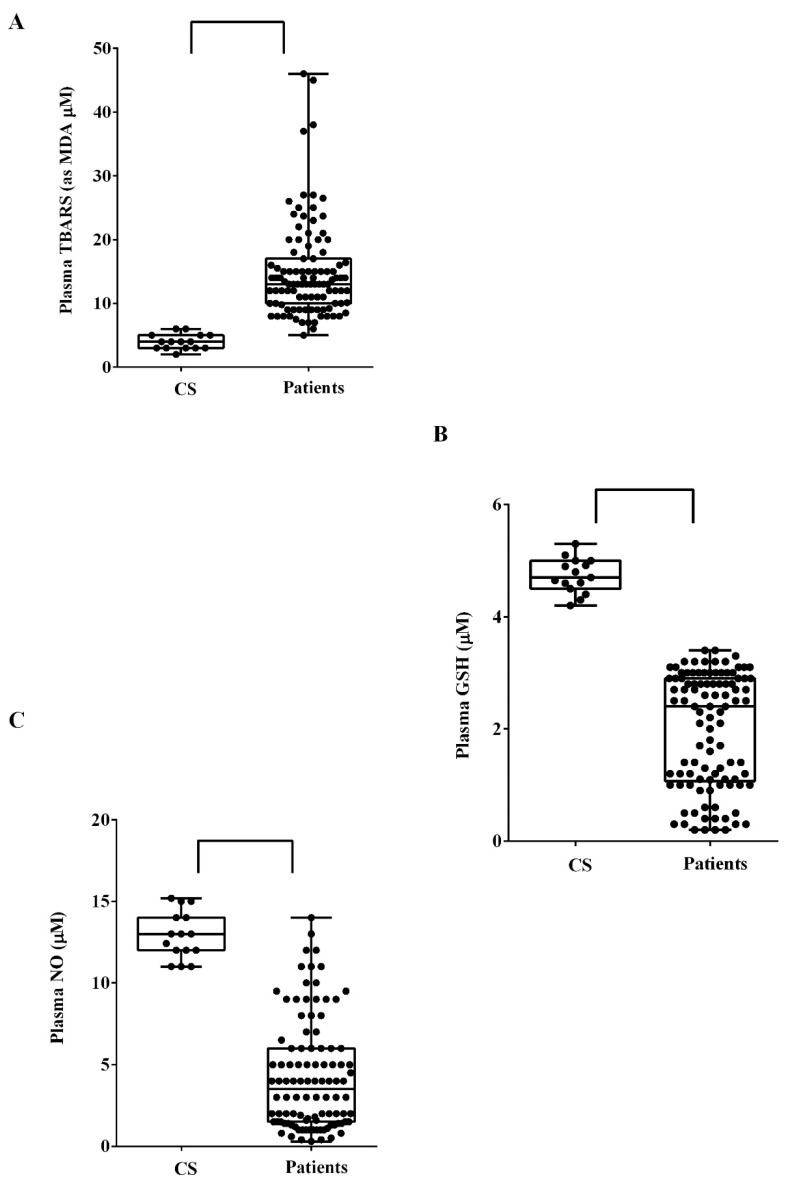
Plasma TBARS (**A**), GSH (**B**) and NO (**C**) levels in ALS patients. CS: control samples. GSH: glutathione; MDA: malonyldialdeide. NO: nitric oxide. Square brackets indicate significance between groups as *p* < 0.05.

**Figure 3 biomedicines-10-00691-f003:**
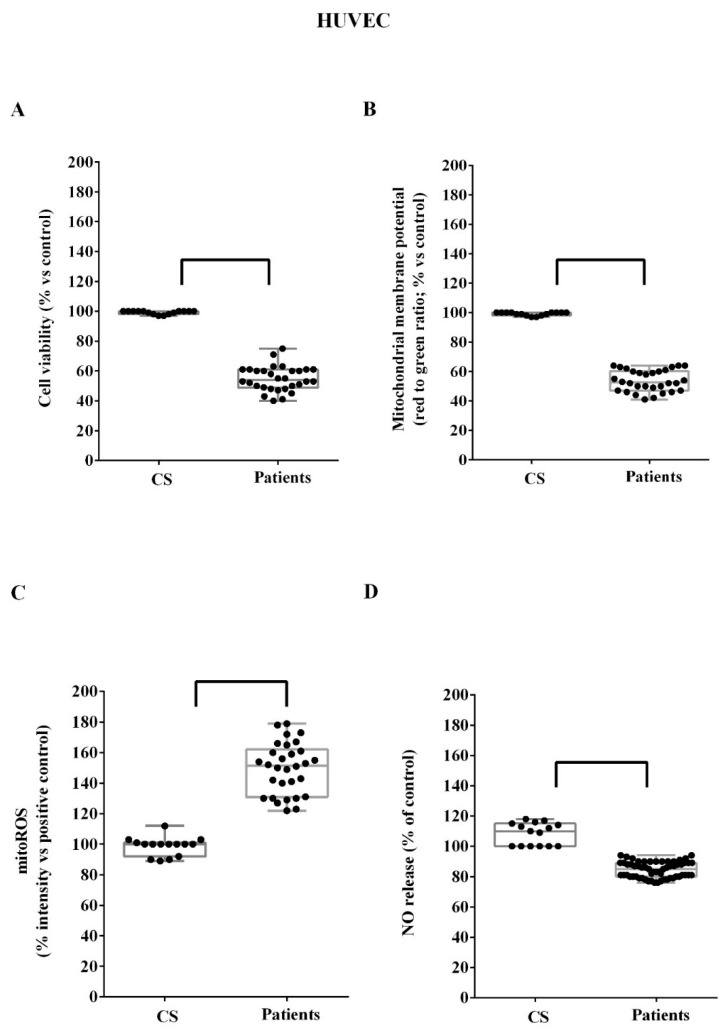
Effects of ALS patient plasma on cell viability (**A**), mitochondrial membrane potential (**B**), mitochondrial reactive oxygen species (mitoROS; (**C**)), and NO release (**D**) in HUVEC. CS: control samples. NO. nitric oxide. Square brackets indicate significance between groups as *p* < 0.05.

**Figure 4 biomedicines-10-00691-f004:**
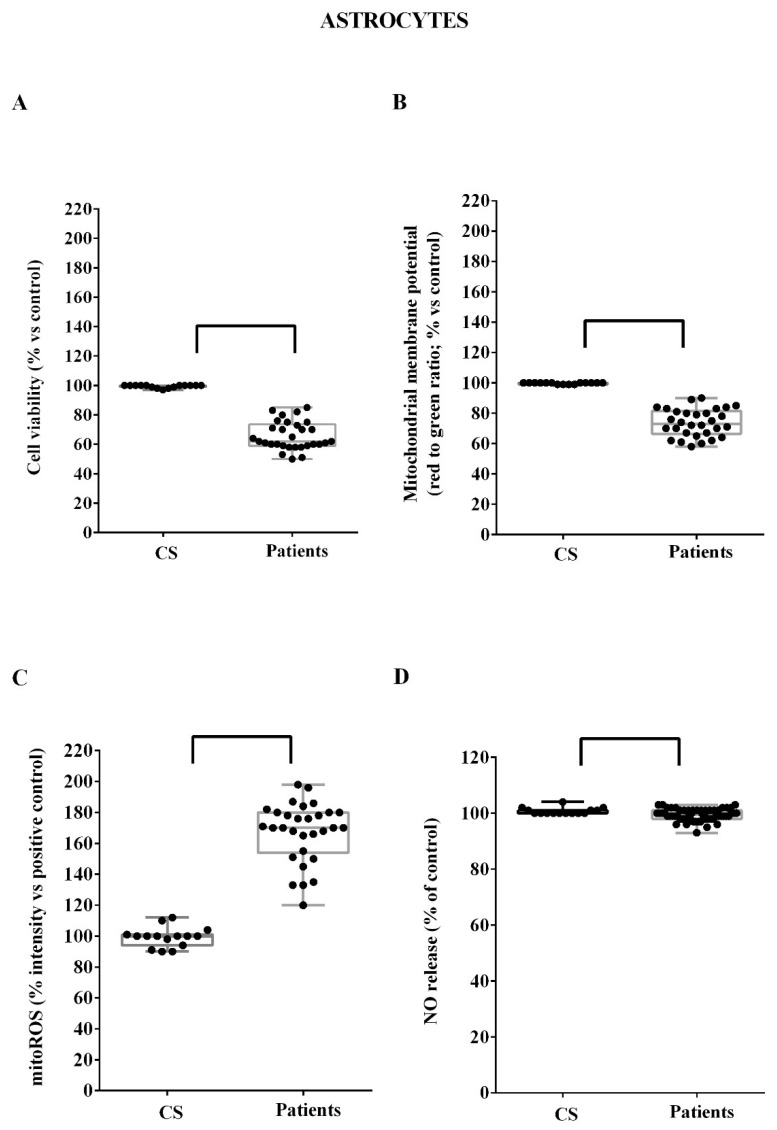
Effects of ALS plasma on cell viability (**A**), mitochondrial membrane potential (**B**), mitochondrial reactive oxygen species (mitoROS, (**C**)) release and nitric oxide (NO) release (**D**), in astrocytes. CS: control samples. Square brackets indicate significance between groups as *p* < 0.05.

**Table 1 biomedicines-10-00691-t001:** Demographic and phenotypic features of ALS patients. Numbers are expressed as number, percentage or median, and interquartile range (IQR) as required. ALSFRS-R: Amyotrophic Lateral Sclerosis Functional Rating Scale—Revised; FVC: forced vital capacity; BMI: body mass index.

Patients’ Features (*n* = 25)	
Sex (male/female)	17 (68%)/8 (32%)
Age	66 (IQR: 59–69)
Phenotype (spinal/bulbar)	18 (72%)/7 (28%)
ALSFRS-R at baseline	40 (IQR: 37–43)
FVC% at baseline	82 (IQR: 57–100)
BMI at baseline	22 (IQR: 20–24)

## Data Availability

The data presented in this study are available on request from the corresponding author.

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
