# Peer review of "The Potential Role of Peripheral Oxidative Stress on the Neurovascular Unit in Amyotrophic Lateral Sclerosis Pathogenesis: A Preliminary Report from Human and In Vitro Evaluations"

_biomedicines, 2022, doi:10.3390/biomedicines10030691_

Round 1
Reviewer 1 Report
In the research article “Role of the neurovascular unit and the redox state in amyotrophic lateral sclerosis pathogenesis: a preliminary report from human and in vitro evaluations” Grossini et. al., have performed a relative quantification of thio-barbituric acid reactive substances (TBARS), glutathione (GSH) and nitric oxide (NO) in the plasma of 25 ALS patients with respect to the healthy controls. Here authors showed increase in the TBARS and decrease in the GSH and NO in the plasma from most of the ALS patients in comparison to controls. Authors, further determined the effect of ALS plasma (from 5 patients) on viability, mitochondrial membrane potential and the release of mitochondrial reactive oxygen species (mitoROS) in the HUVEC and astrocytes and NO release in HUVEC. As, the follow up of patients is still going on, so some interesting results are expected in continuation of this study. However, at present, authors should address the following points.
- Quantification of nitric oxide (NO) release from astrocytes treated with ALS patient and healthy control plasma.
- Increase the number of plasma samples from ALS patients for the in vitro experiments (only 5 out of 25 patients was used in the depicted results), which will increase the confidence for the in vitro results, specifically for the NO release from HUVEC and astrocytes.
Plagiarism percentage - not checked by the reviewer.
Author Response
From:
Letizia Mazzini
Department of Neurology and ALS Centre,
University of Piemonte Orientale;
Maggiore della Carità Hospital, Corso Mazzini 18, 28100 Novara;
Tel: +39-0321-3733962;
E-mail: Letizia.mazzini@uniupo.it
To:
Editor-in-Chief
Biomedicines
March, 18th, 2022
Subject: Revision and resubmission of manuscript “Role of the neurovascular unit and the redox state in amyotrophic lateral sclerosis pathogenesis: a preliminary report from human and in vitro evaluations.”
Dear Editor,
Thank you for your letter and the opportunity to present our manuscript entitled “Role of the neurovascular unit and the redox state in amyotrophic lateral sclerosis pathogenesis: a preliminary report from human and in vitro evaluations” on your Journal. The suggestions offered by the reviewer are helpful, and we also appreciate your interest on the paper.
In this letter, we have included the reviewer comments, and our replies. We also checked the plagiarism and it results less than 5%.
We hope that the revised paper will be suitable for publication in Biomedicines.
We thank you for your continued interest in our research.
Sincerely,
Fabiola De Marchi
------------------------------------------------------------------------------------------------------------------------------------------
Response to reviewer 1:
We would like to thank the reviewer for careful and thorough reading our paper. Our response follows the reviewer comments, which are in are in italics.
Comment 1 - Reviewer #1: Content:
Quantification of nitric oxide (NO) release from astrocytes treated with ALS patient and healthy control plasma.
Reply to reviewer: we agree with the reviewer and have performed new experiments about NO release by astrocytes. As described in the revised manuscript, ALS plasma caused a reduction of NO release by astrocytes. We have added new data in Figure 4 (panel D), Results (page 10, lines 266, 267) and discussed the results obtained in the Discussion section (page 12, lines 343-347).
Comment 2 - Reviewer #1: Content:
Increase the number of plasma samples from ALS patients for the in vitro experiments (only 5 out of 25 patients was used in the depicted results), which will increase the confidence for the in vitro results, specifically for the NO release from HUVEC and astrocytes.
Reply to reviewer: we agree with the reviewer about the fact that increasing the number of plasma samples for the in vitro experiments would increase the confidence for the results. By this way we performed new experiments about NO release in both HUVEC and astrocytes by the use of 5 more plasma samples (Methods, page 3, lines 139, 140; Results, page 8, lines 252, 253).
Reviewer 2 Report
The paper presented by Grossini et al entitled " Role of the neurovascular unit and the redox state in amyotrophic lateral sclerosis pathogenesis: a preliminary report from human and in vitro evaluations" investigates (1) the redox status of ALS plasma samples and (2) the effects of ALS plasma on survival and oxidative stress on endothelial and astrocyte cells.
This is an interesting preliminary report, very simple, describing the potential role of oxidative stress in plasma on endothelial cells and astrocytes. However, I would suggest few amendements/questions before publications.
First regarding the title, I think according to the data presented, the authors should rethink a little about it as I don't think the "role of the neurovascular unit" has been tested but instead they investigated "the potential role of peripheral oxydative stress on the neurovascular unit".
See below the details of my comments.
General comment: I would suggest to the authors to present the data as an average ±SD and performed the statistical analysis using these averages, instead of comparing Control versus individual patient replicates. Individual values can be shown as dots inside the box plots. Non parametric test were used for this paper, and seems to be the correct approach. However, I suspect that the statistical analysis done in the current paper is comparing the mean of the 5 controls versus mean of replicates of each ALS patients, using multiple time the Mann-Whitney test. If I'm correct, I don't think that is the correct strategy. Please, represent the stat on each box plot, and stat comparing the mean of control vs mean of ALS values. This reviewer understands that the variability between ALS patients (as it is an heterogeneous group to start with, despite precocious clinical recruitment strategy) may remove some statistical significance. Unless the authors have a good raison or a good strategy to stratify the patients in different subgroup etc, I think the most classic way to compare the data (means vs means) will be the most appropriate strategy. If the authors persist to compare each patient replicates with control average, then a Kruskal-Wallis test should be applied at the very least (though still not fully correct).
Would it be possible to insert a table summarising the clinical data for all subject studied? As a plus, the authors could also indicate which patient plasma samples were used for the cell culture treatment it would be great.
Figure 1: try to avoid distortion. May be good to start the schema with the serum to be added to the insert, then make a zoom in the inset to show where the cells are plated and where the plasma is added (in other word, same figure but invert it)
Figure 2: not clear what are the different letter a, b,c etc and the numbers on the x -axes, I'm guessing different ALS patients. Why not showing instead the box plot only and add the individual values with dot for each patients (without showing the replicates)
Figure 3 and 4: The same comment for data representation and analysis. Comparison CS vs ALS is sufficient as all values are normalised to untreated., or if comparison C vs CS vs ALS then Kruskal-Wallis test should be performed instead of multiple Mann-Whitney Also specified in these figures which cell types was studied (Endothelial cells or astrocytes) to facilitate the reading. Why NO was not investigated for astrocyte treatment?
Discussion: the authors should discuss a little bit further the limits of their study, example: whole plasma studied, cannot target a specific circulating element(s) that could affect the oxydative stress and toxicity toward endothelial cells and astrocytes.
Would the authors have the possibility to test artificial BBB/BSCB and the capacity of the plasma to affect the permeability of this artificial BBB/BSCB?
Would some ROS scavenger reverse the ALS plasma effect on endothelial cells and/or astrocytes?
Author Response
From:
Letizia Mazzini
Department of Neurology and ALS Centre,
University of Piemonte Orientale;
Maggiore della Carità Hospital, Corso Mazzini 18, 28100 Novara;
Tel: +39-0321-3733962;
E-mail: Letizia.mazzini@uniupo.it
To:
Editor-in-Chief
Biomedicines
March, 18th, 2022
Subject: Revision and resubmission of manuscript “Role of the neurovascular unit and the redox state in amyotrophic lateral sclerosis pathogenesis: a preliminary report from human and in vitro evaluations.”
Dear Editor,
Thank you for your letter and the opportunity to present our manuscript entitled “Role of the neurovascular unit and the redox state in amyotrophic lateral sclerosis pathogenesis: a preliminary report from human and in vitro evaluations” on your Journal. The suggestions offered by the reviewer are helpful, and we also appreciate your interest on the paper.
In this letter, we have included the reviewer comments, and our replies. We also checked the plagiarism and it results less than 5%.
We hope that the revised paper will be suitable for publication in Biomedicines.
We thank you for your continued interest in our research.
Sincerely,
Fabiola De Marchi
------------------------------------------------------------------------------------------------------------------------------------------
Response to reviewer 2:
We would like to thank the reviewer for careful and thorough reading our paper. Our response follows the reviewer comments, which are in are in italics.
Comment 1 - Reviewer #2: Content:
First regarding the title, I think according to the data presented, the authors should rethink a little about it as I don't think the "role of the neurovascular unit" has been tested but instead they investigated "the potential role of peripheral oxidative stress on the neurovascular unit".
Reply to reviewer: thanks for your advice. We changed the title as the reviewer suggested. The new title is “The potential role of peripheral oxidative stress on the neuro-vascular unit in amyotrophic lateral sclerosis pathogenesis: a preliminary report from human and in vitro evaluations”.
Comment 2 - Reviewer #2: Content:
I would suggest to the authors to present the data as an average ±SD and performed the statistical analysis using these averages, instead of comparing Control versus individual patient replicates. Individual values can be shown as dots inside the box plots. Non parametric test were used for this paper, and seems to be the correct approach. However, I suspect that the statistical analysis done in the current paper is comparing the mean of the 5 controls versus mean of replicates of each ALS patients, using multiple time the Mann-Whitney test. If I'm correct, I don't think that is the correct strategy. Please, represent the stat on each box plot, and stat comparing the mean of control vs mean of ALS values. This reviewer understands that the variability between ALS patients (as it is an heterogeneous group to start with, despite precocious clinical recruitment strategy) may remove some statistical significance. Unless the authors have a good raison or a good strategy to stratify the patients in different subgroup etc, I think the most classic way to compare the data (means vs means) will be the most appropriate strategy. If the authors persist to compare each patient replicates with control average, then a Kruskal-Wallis test should be applied at the very least (though still not fully correct).
Reply to reviewer: As suggested, we compared the two groups (controls vs ALS patients) using the Mann Whitney test. We used non parametric test because data are not normally distributed and the samples are small. We showed in the graphs now represented as box plots, the individual data.
Comment 3 - Reviewer #2: Content:
Would it be possible to insert a table summarising the clinical data for all subject studied? As a plus, the authors could also indicate which patient plasma samples were used for the cell culture treatment it would be great.
Reply to reviewer: thanks for the possibility to add a Table and avoiding a redundant paragraph (see in the Manuscript as Table 1). We randomly selected the plasma samples for the cell culture treatment (we specified this point in the text, page 3 lines 138 and 140).
Comment 4 - Reviewer #2: Content:
Figure 1: try to avoid distortion. May be good to start the schema with the serum to be added to the insert, then make a zoom in the inset to show where the cells are plated and where the plasma is added (in other word, same figure but invert it)
Figure 2: not clear what are the different letter a, b,c etc and the numbers on the x -axes, I'm guessing different ALS patients. Why not showing instead the box plot only and add the individual values with dot for each patients (without showing the replicates)
Figure 3 and 4: The same comment for data representation and analysis. Comparison CS vs ALS is sufficient as all values are normalised to untreated., or if comparison C vs CS vs ALS then Kruskal-Wallis test should be performed instead of multiple Mann-Whitney Also specified in these figures which cell types was studied (Endothelial cells or astrocytes) to facilitate the reading. Why NO was not investigated for astrocyte treatment?
Reply to reviewer: thanks for the advice. In detail:
1) Figure 1: we edited it
2) Figure 2: We deleted the single columns from figure 2 and shown various patients as single dots, now
3) Figure 3 and 4: We have deleted various columns from figures 3 and 4 and substituted them with box plot with dots for single values. We have also specified in figure the cell line (HUVEC or astrocytes) and have also added NO from astrocytes, now.
Comment 5 - Reviewer #2: Content:
Discussion: the authors should discuss a little bit further the limits of their study, example: whole plasma studied, cannot target a specific circulating element(s) that could affect the oxydative stress and toxicity toward endothelial cells and astrocytes.
Reply to reviewer: we thank the reviewer for the possibility to improve the Discussion paragraph of our paper. Mainly, we added some points as limitations (from line 348 to the end).
Comment 6 - Reviewer #2: Content:
Would the authors have the possibility to test artificial BBB/BSCB and the capacity of the plasma to affect the permeability of this artificial BBB/BSCB? Would some ROS scavenger reverse the ALS plasma effect on endothelial cells and/or astrocytes?
Reply to reviewer: we have addressed the points by adding more paragraphs in the Discussion section (page 12, lines 358-374).
Round 2
Reviewer 1 Report
In the revised version of the manuscript, authors have addressed the queries raised by reviewer's. Therefore, this article should be consider for the publication.